# Safety Concern and Regulatory Status of Chemicals Used in Cosmetics and Personal Care Products

**Manthan Kaushik** [1], **Uzma Farooq** [2], **Mohd Shoab Ali** [2], **Mohammad Javed Ansari** [3], **Zeenat Iqbal** [2,*] **and Mohd Aamir Mirza** [2,*]

1. School of Pharmaceutical Education & Research, Jamia Hamdard, New Delhi 110062, India; manthankaushik.2000@gmail.com
2. Department of Pharmaceutics, School of Pharmaceutical Education & Research, Jamia Hamdard, New Delhi 110062, India; uzma411@gmail.com (U.F.); mohdshoebali@gmail.com (M.S.A.)
3. Department of Pharmaceutics, College of Pharmacy, Prince Sattam Bin Abdulaziz University, Alkharj 16278, Saudi Arabia; javedpharma@gmail.com
* Correspondence: zeenatiqbal@jamiahamdard.ac.in (Z.I.); aamir.jhu07@gmail.com (M.A.M.); Tel.: +91-9811733016 (Z.I.)

**Abstract:** Cosmetics and personal care products (PCPs) are a few of the most commonly used products across the globe with a whopping market share of approximately USD 500 billion. These products are used for cleansing purposes and for improving the quality and beauty of the face, hair, and skin. There are many chemical substances involved in the manufacturing of cosmetics and PCPs. These chemical substances incorporated in cosmetics or PCPs are crucial to develop high-quality products with superior appearance, applicability, and stability; however, excessive use of such chemicals in cosmetics and PCPs has become a safety concern as many of these are reported to cause severe health complications. Overuse of cosmetics and PCPs with hazardous material should be minimized, especially by pregnant women and children. Gynecologists advise pregnant women not to use cosmetics and PCPs with hazardous chemicals. The implementation of a lawful framework is crucial to establish the safety of cosmetics and PCPs. Cosmetic companies/industries must be strictly regulated and made compliant to the guidelines in order to protect human health and minimize safety concerns. In this review, hazardous chemicals incorporated in the personal care products/cosmetics and their related risk and health complications have been discussed in detail. Additionally, regulatory status and clinical trials of chemical substances that involve toxicity and causing severe complications have also been discussed.

**Keywords:** personal care products; harmful chemicals; hair dyes; 1,4-dioxane; parabens; cosmetics; regulation; cancer

## 1. Introduction

According to Section 3(a) of the Drugs and Cosmetics Act, 1940 "Cosmetic can be explained as a product anticipated for coating, gushed, dispersed or sprayed on, or instigate into, or else dermally used on the humans or any part of the things just mentioned for cleansing, embellishing, enhancing attractiveness, or changing the looks, and includes any product intended for a component of cosmetic". The definitions of cosmetics as per different stringent regulatory agencies are presented in Table 1 [1,2]. It can be said that beauty products are the ones that help to clean and beautify human skin/hair or appearance. These products are also intended to repair damaged skin.

After globalization in 1980s, the cosmetic/PCPs industry boomed globally. These products are helping millions of people to improve their appearance, cleanliness, hydration, and moisturize the body. Some of the constituents in these products are toxic for human use and may cause serious health concerns [3,4]. Regularly, every culture of the world is known to apply their required cosmetic/PCPs that are mainly used for protection, cleaning,

beauty, hygiene, and well-being [5]. Approximately 60–80% of pregnant women use these skincare products, hair products, and facial makeovers for the long term. As we know, the chemical components of these cosmetic products are dermally absorbed and have a chance of crossing the skin and entering the systemic circulation [6]. These products include hair care products, oral cavity products (toothpaste and mouthwashes, etc.), and mucosal products used on the scalp, eyes, face, and lips [7]. The dermal products series are classified as leave-on products applicable for their limited period of use or intended for cleansing/wash-off products. On the basis of time/duration of application, some are applied regularly (moisturizer, lip care, eye makeup, etc.), seasonally (sunscreen and winter lotion), and occasionally (hair color dye). Other categories are related to consumer interest such as daycare cream/moisturizer plus ultra-violet protective products [8,9].

**Table 1.** Definition of cosmetics according to regulatory agencies.

| Regulatory Agencies | Definition of Cosmetics |
|---|---|
| USFDA | "A product (except pure soap) that is used to be applied dermally for cleansing, whitening, and beautifying purposes" |
| CDSCO (India) | "An article anticipated for coating, pouring, and sprinkling or spraying on, or incorporated into, or otherwise applied on the human skin or any part thereof for cleansing, embellishing, enhancing attractiveness, or improve appearance, and includes any article used for a component of cosmetic." |

When these cosmetic products penetrate dermally after getting absorbed, they can cause severe complications such as endocrine disorders, cancer, neurological disorder, reproductive disorders, etc. There are more than 13,000 synthetic and other industrial chemicals that are incorporated in cosmetic products; however, only 20% of these chemicals are perceived as safe, and illustrated by the USFDA [10].

The threshold of toxicology concern (TTC) study of chemical compounds of cosmetics/PCPs has limited absorption data as functional barriers work on the stratum corneum. Quantitative study data of chemicals that have crossed the skin barriers are determined through in vitro studies. Absorption data of chemical constituents are expressed with the coefficient of dermal permeation (kp) and designated as steady-state flux via a concentration gradient [11,12].

Based on the report of the breast cancer fund, an average woman in the US applies more than twelve cosmetic products daily while men use six products. More than 10,000 chemicals are used in the manufacturing of these cosmetic products and ≤20% of chemicals or constituents are reported to be safe [13]. These active chemicals can transdermally penetrate the systemic circulation, which can adversely affect human health. High frequency and duration of application could enhance the risk of cancer, endocrine disorders, neurotoxicity, and reproductive disorders. Following the regulation for safe cosmetics/PCP, the United States has banned 11 chemicals while European Union has banned 1328 chemicals from use in cosmetics [14].

A large group of PCP such as skin whitening products, creams, and body oils contain around 9% of hydroquinone which is used in 30 times higher concentration than the maximum limit (0.3%). It can cause allergic reactions to the skin, ocular disorders, genetic disorders, and even cancer [15].

The high demand for cosmetics/PCP worldwide at all age groups has also elevated awareness associated with safety issues. In this review, a focus has been made on chemicals found in cosmetics/PCPs that are toxic and harmful to human health. This paper also highlights the regulatory issues and safety assessment of chemical ingredients and their related clinical trials. The search criteria of the key factors required in the given review have been mentioned in Figure 1.

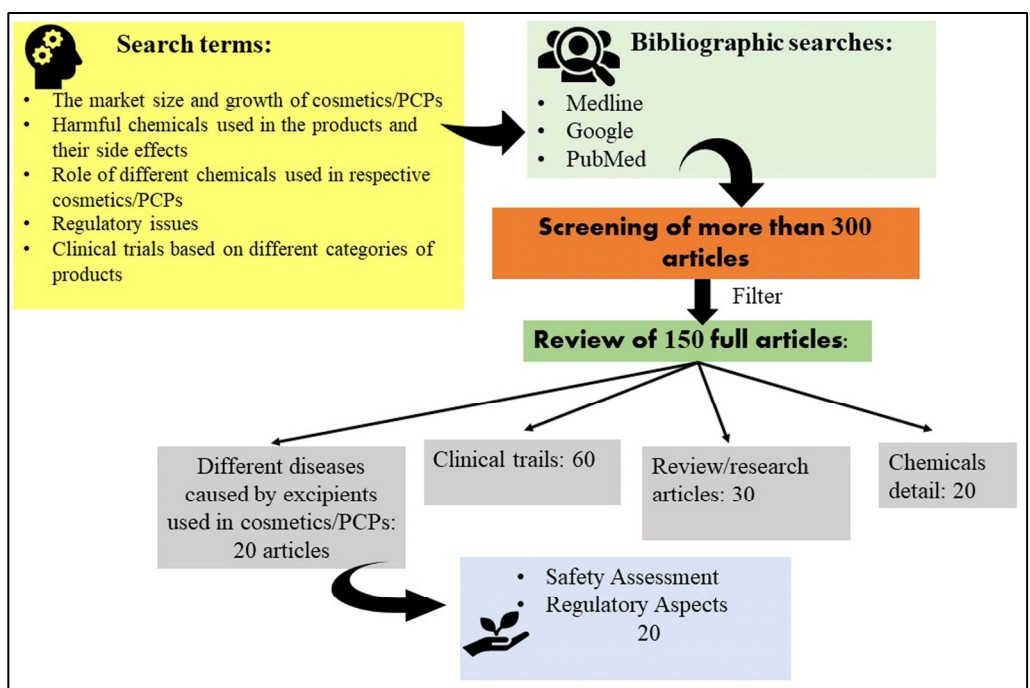

**Figure 1.** Searching criteria and filtration of literature.

## 1.1. Range of Personal Care or Topical Products

Personal care products include face wash, moisturizer, cotton swabs, eyeliner, lip balm, deodorant, cologne, sunscreen, etc. These are applied to improve or maintain the physical appearance of the user [16]. These products are either cleansers or dermally applied cream or lotion (sunscreen, moisturizer, etc.,) for different purposes to clean or to protect the skin from sunburn, dryness, and dust particles [17]. Additionally, they are used for facial makeovers are intended to beautify the look. In the modern lifestyle, consumers are diverting toward natural ingredient-based products with limited toxicity of natural compounds. However, herbal cosmetics also contain other synthetic chemicals such as parabens, artificial colorants, stabilizers, and emulsifiers, etc., [18]. These are not safe for humans and can cause serious health issues. Chemicals such as phenol, triclosan, parabens, ether could cause endocrine disorders, neurological disorders, and risk of other diseases [19]. The cosmetic industry is still growing and has some keys drivers for the growth areas.

## 1.2. Market Size and Growth

In 2019, world's beauty care industry had a capital of USD 532 billion [20]. In 2019, the biggest consumer was the United States (US) which occupies 20% of the overall market size; China occupied the second position and third position by Japan with a market percentage of 13% and 8%, respectively. The cosmetic industry is expected to reach USD 800 billion by the year 2025 with a CAGR (compound annual growth rate) of 5–7% [21]. The cosmetic industry is the largest sector under the beauty care industry. Top leading companies are L'Oreal, Unilever, Estee Lauder, Coty, Proctor, and gamble. In the year 2018–2019, the Indian cosmetic market was valued at USD 6.5 billion. The top leading companies in India are Boutique, Lotus, Patanjali, Coach, Estee Lauder Companies Inc. [22] According to recent data, the market growth size of cosmetic industries increased through the increment in consumer's income and urbanization, increment in consumer demand, lifestyle modification, rising interest in male grooming, and also increased demand for natural or bio-based products [23].

### 1.3. Harmful Chemicals Present in the Products

Despite all the positivitiess of cosmetics, we need to focus on safety measures of these hazards as it is a matter of human health. Chemicals and other ingredients present in these products can be hazardous [24]. A lot of these chemicals used as preservatives, UV filters, color additives, etc., can cause cancer, mutation, developmental and reproductive toxicity, endocrine disruption, etc., [25]. Most of the cosmetic products are applied on the skin, which results in the toxic/harmful chemicals penetrating the stratum corneum barrier and reaching the dermis of the human skin, and may further reach the systemic circulation [26]. Hence, it is capable of disrupting fertility during pregnancy and causing female reproductive disorders as shown in Figure 2. All the ingredients in the beauty products meet certain regulatory obligations, but some ingredients are allowed at low concentrations due to their toxicity at higher concentrations. Facial wash or cleansers might not be ideally safe for regular use on or around the delicate and specialized dermal layer of eyelids. These face wash/cleansers contain sodium lauryl sulfates that remove delicate natural oil/moisture on eyelids by evaporation [27]. Globally, lip care products are high-selling cosmetic products in the modern era used to enhance self-esteem and make attractiveness, and moisturize the lips. In lip care products, chemicals such as dye, mineral mica, colorant, and shiner are incorporated during manufacturing to develop high-quality products and boost its effect. These chemicals contain trace metals such as antimony, arsenic, copper, etc., [28] can cause severe adverse effects. Regular long-term use of lip care cosmetic reveal that when these trace metals are used in high quantity they can penetrate the systemic blood circulation causing damage of various organs [29].

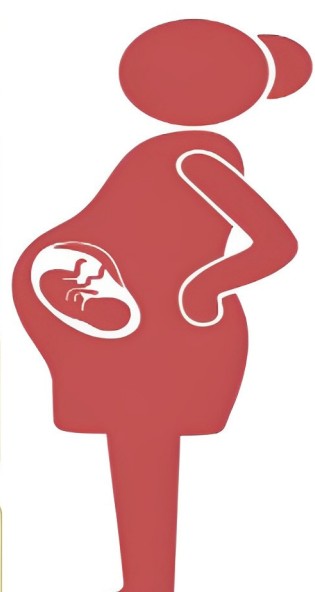

**Blush:** Sixteen chemical are involved in production of blush in which parabens are also used
**Eye shadow:** Twenty six chemicals are used in production of it including polyethylene terephthalate. linked to cause infertility, cancer, endocrine disruption and organ damage.
**Eyeliner/mascara** also contain polyethylene terephthalate and polyethylene propylparaben reponsible for possible carcinogen endocrine disruption and infertility
**Foundation:** approximatly twenty four chemicals are incorporated in the preparation including polymethylmethacrylate that adversely effect on immune system, allergic reaction and risk of cancer.

**Skincare/bodycare:** It is avoided during pregnancy as It contain parabens, mineral oils, sulphates, PEGs, phthalates, perfumes and alcohol.

**Lipstick:** it contain polymenthyl methacrylate or polyethylene terephthalate that cause cancer, infertility, hormonal imbalance and damage to organs.
During pregnancy, It Is safe to avoid as It contain synthetic colors and often contaminated with heavy metals.

**Nail polish/varnish:** it contain phthalate, toluene, and formaldehyde. It contain 31 chemicals that causes hormonal imbalance, fertility problems, cancer and also creat problems in the development of foetus. it is safe to avoid during pregnancy

**Sunscreen:** in preganancy, it is advise to avoid oxybenzone and synthethic oestrogen as it might reduce birth weight. it also cause hormonal disruption and foetus developmental complication

**Perfumes:** cocktail of 100 chemicals like benzaldehyde and toluene that might cause organ dysfunction, cancer, hormone dysruption, male reproductive organ disorder and development of male reproductive organ in foetus.

**Toothpast:** toothproduct containing tricosian, sodiumlauryl sulphate and artificail sweetners are responsible for reproductive disorders.

**Hair dyes:** Octinoxate and isophthalate can adversely cause allergies, hormonal disorder, infection or irritation to eyes, nose and throat.

**Shampoo:** Approx. 15 chemicals are involved in the manufacturing of shampoo. SLS, PEG, Methylisothiazoline are adversly cause neurological damage in fetus, eye Infection/irritation/damage.

**Figure 2.** Cosmetic products/PCP which are routinely used by young women and pregnant women possess hazardous chemicals that can cause severe complications.

## 2. Major Health Concerns

### 2.1. Cancer

Cancer is a disease in which aberrant cells divide with no control and conquer nearby cells and tissues. Chemicals that are used in beauty care products either directly show carcinogenic effects or have a risk of causing cancer [14]. Fluorene and phenanthrene (group 3) along with naphthalene (group 2B), chrysene (group 2A), and titanium dioxide (group 2B) are carcinogenic agents capable of inducing lung cancer; however, other chemicals such as naphthalene, N-nitrosodiethanolamine (group 2A) can develop liver cancer. Similarly, chemicals such as butylatedhydroxyanisole (group 2A) can cause stomach cancer [30]. Formaldehyde and formaldehyde-releasing preservatives (group1) can cause leukemia (blood cancer). Some chemicals such as avobenzone, hydroquinone, and PABA present in the sunscreens are not carcinogenic but can increase the risk of having skin cancer, while coal tar (group 1) is carcinogenic that can cause skin cancer as shown in Figure 3 [31]. 1,4-Dioxane (group 2A) present in hair products can cause breast cancer, while p-phenylenediamine present in hair dyes can increase the risk of skin cancer [28]. Arsenic (group 1) can cause bladder cancer, and acrylates (group 2B) in nail products can cause colorectal cancer [29].

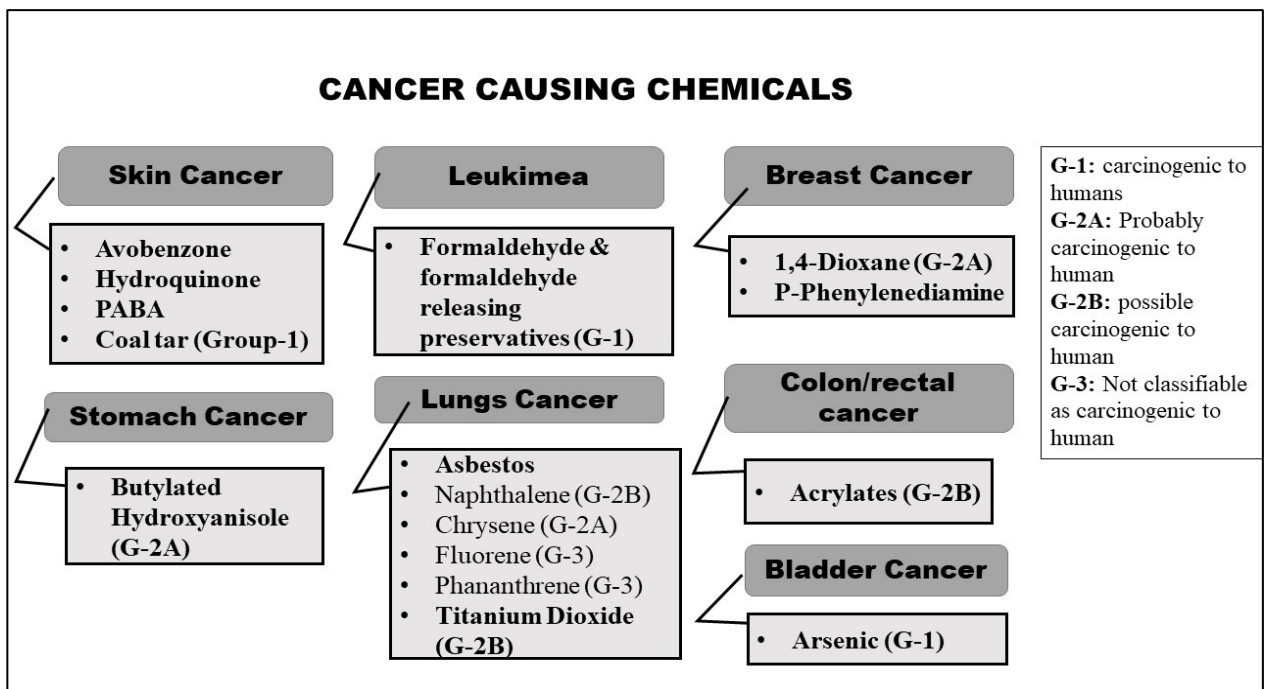

**Figure 3.** Range of toxic chemicals used in cosmetic/personal care products which causes cancer.

### 2.2. Organ System Toxicity

Harmful ingredients present in beauty care products can accumulate in our bodies and might cause organ system toxicity [32]. As can be seen in Table 2 and Figure 4, many ingredients can cause various types of organ system toxicity. Substances such as methylparaben, propylparaben, butylated hydroxy-toluene, acrylates, and octinoxate (present in shampoos, sunscreen, and nail products) can cause developmental toxicity [33]. Substances such as p-phenylenediamine, coal tar, and hydroquinone can cause skin toxicity. Butylated hydroxy-anisole and cadmium can cause renal toxicity [34]. Methylparaben, propylparaben, acrylates, and octinoxate can also cause reproductive toxicity. Ingredients such as ether, butylated hydroxy-anisole, and methylisothiazolinone (used in hair products) possess risk of causing lung toxicity while benzophenone can cause liver toxicity [35].

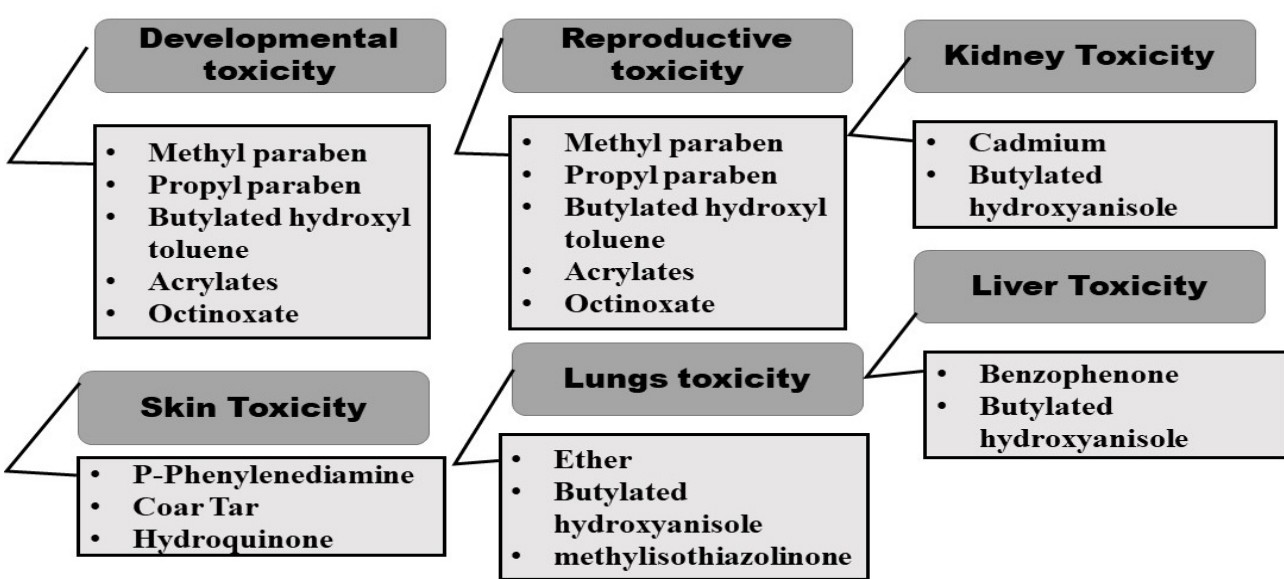

**Figure 4.** Range of toxic chemicals used in cosmetic/personal care products causes organs/tissues toxicity.

*2.3. Endocrine Disruption*

Endocrine disruptors are chemical substances that are capable of disrupting hormonal signals thus affecting the reproductive and nervous systems. Various grades of phthalate can prevent the synthesis of testosterone, while antimicrobial agents (triclosan) cause hypothyroidism [36]. We can generalize endocrine disruptors present in beauty products in two parts namely reproductive health endocrine disruptor and thyroid endocrine disruptor, as shown in Figure 5. Ingredients such as avobenzone, butylparaben, methylparaben, octinoxate, and butylatedhydroxyanisole (present in sunscreen and hair products) are reproductive health endocrine disruptors and can cause fertility abnormalities, skewed sex ratio, and in females can also cause menstrual problems [37]. Chemicals such as PABA, padimate O, homosalate, triclosan, octinoxate, and resorcinol can disrupt the thyroid hormone and can cause hyperthyroidism or hypothyroidism. Generally, there are no metabolic disruptors present in any beauty care products [38]. In a a study on the examination of urine samples of a specified group of adults, in more than 96% of the samples methyl/propylparaben was obtained. Several other studies have also reported the high concentration of parabens in various products, dermal penetrability, and its toxic effects [39].

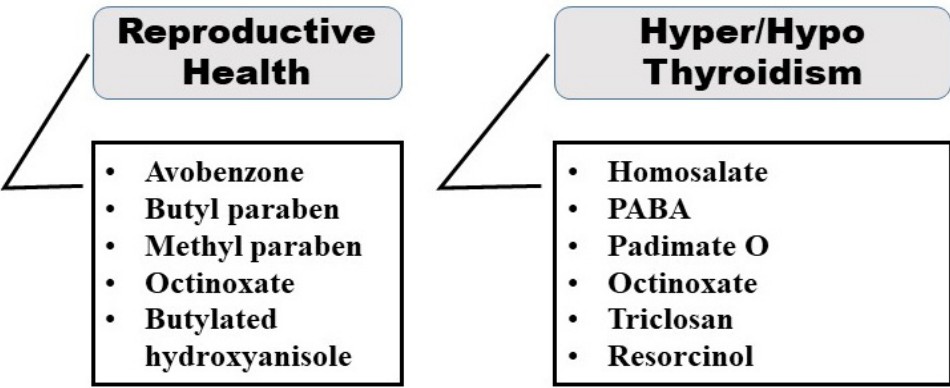

**Figure 5.** Range of chemicals used in cosmetic/PCPs can cause endocrine disruption.

### 2.4. Allergies, Irritation, and Asthma

Allergies are the body's immune system shows hypersensitive response to a foreign substance (harmful or harmless). Different people present different kinds of reactions to similar allergens. Skin allergies are one of the most common side effects of cosmetic or beauty care products. Allergic reactions could also be occurring by some chemicals that are common in cosmetics manufacturing, like alcohol (in shampoos, lotions), isothiazolinones (preservatives added in shampoos and body wash), butylated hydroxytoluene (preservative in hair products), and ether (as an impurity in hair products) [36].

### 2.5. Other (Cellular and Neurological Damage, Ochronosis, Sensitization)

Other health issue that can be caused by using beauty care products are ochronosis caused by hydroquinone (present in skin lightener and facial moisturizer), chemicals such as oxybenzone (present in sunscreen), diethanolamine (present in soap and shampoos), padimate-O (present in sunscreen), coal tar (present in shampoos and hair dyes) can cause bioaccumulation [40]. P-phenylenediamine and benzalkonium chloride are mutagenic, acrylates can cause cellular and neurological damage, mercury present in eye area products can cause mercury poisoning, and methylisothiazolinone (present in shampoos as preservatives) can cause neuroticism as mentioned in Figure 6 [23].

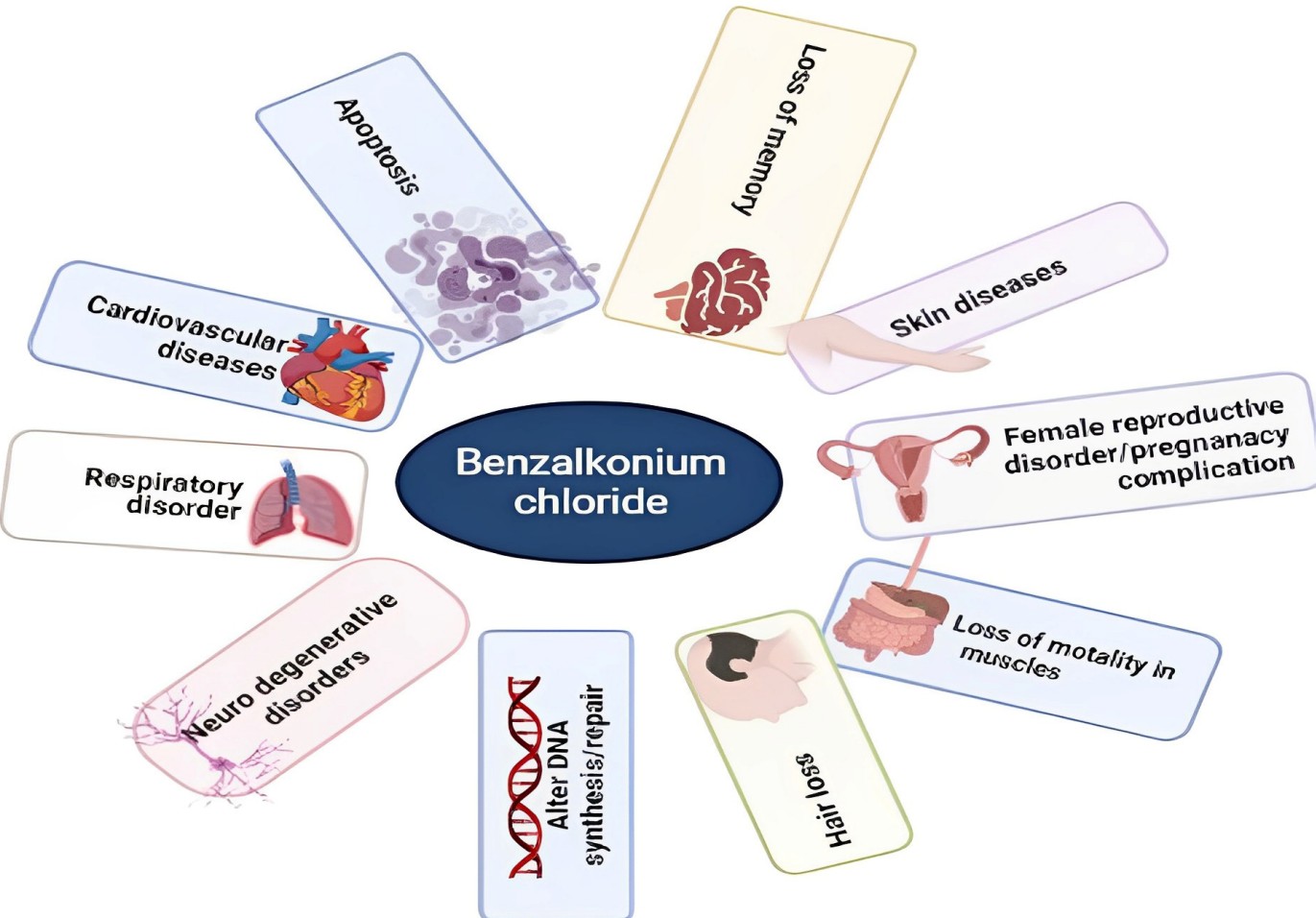

**Figure 6.** Severe complications caused by benzalkonium chloride commonly used in cosmetic/personal care products.

### 3. Regulatory Issues

In USA, cosmetics regulation comes under FDA and is authorized by the Federal Food, Drug, and Cosmetic Act (FD & C Act) and Fair Packaging and Labeling Act (FPLA). Except

for color additives, all the manufacturing items and excipients do not require any FDA premarket approval [31]. Manufacturers have the legal authority to guarantee the safety of products [41]. Based on the Safe Cosmetic and Personal Care Product Act, manufacturing brand holders in the USA have to register and renew annually and deposit actual data of excipients used in cosmetic products. In 2013, a subcommittee of workforce protection was referred by the secretary of health and human services [42,43]. Complete information on various chemicals used in cosmetics/personal care products including their major health concern and regulatory status are shown in Table 2.

## 4. Safety-Based Global Clinical Trials of Personal Care Products

In clinical trials, research is carried out on people to study the effect of new tests and treatments in order to observe and evaluate their effects on the human body. These trials include a test of new drugs, surgical treatment, biological products, etc., on people volunteering to take part in these clinical trials [44]. These clinical trials first have to be approved by the respective authorities of that country. Clinical trials have four phases: Phase (1) when the trial is carried out on healthy volunteers; phase (2) when the trial is carried out on a small population of the diseased patient; in the third and fourth phases, the test is approved for human use on a large populationto check its effect on the human body in a longer time frame [45]. Global data of clinical trials of cosmetics/PCPs are shown in Table 3a–c.

**Table 2.** Range of chemicals used in cosmetics/personal care products and their major health concern and regulatory status.

| S.No. | Category and Name of the Ingredients, Types of Cosmetic Products and Functions | The Amount Used in PCP, Other Pharmaceutical Products and Major Health Concerns | References |
|---|---|---|---|
| 1. | Acrylates: Ethyl acrylate, ethyl methacrylate and methyl methacrylate <br> **Used in**: Artificial nail products <br> **Functions**: Acrylic nails, nail modifying polish | **The amount used in PCP**: No regulation prohibits the use of 100% concentration of acrylates in cosmetic products, but acrylates in cosmetics are banned in some states of the USA <br> **Major health concerns**: <br> Cancer, toxic effect on various organs, cellular damage, and neurological disruption. Developmental and reproductive, toxicity, cellular and neurological injuries <br> **Regulatory Status**: <br> • Used as an active drug. Monograph available in USP 29 but not in BP 2017, EP 7.0, JP (17th edition) and IP 2018 <br> • USFDA and Health Canada accept up to 6% in their products | [46] |
| 2. | Aromatic amine: Diazolidinyl urea, p-phenylenediamine <br> **Used in**: Permanent hair dyes <br> **Function**: Hair colorant | **The amount used in PCP**: 0.5% $w/w$ (US) <br> 6% $w/w$ in hair dyes <br> **Major health concerns**: Skin sensitization, cancer, mutagenicity, and organ system toxicity | [47–49] |
| 3. | Asbestos: Chrysotile, Amosite, crocidolite, tremolite, actinolite and anthophylite <br> **Used in**: Baby powder, lotion, foundation | **The amount limit**: There is no federal law in the USA requiring Talscum to be asbestos free <br> **Major health concerns**: Cancer, irritation, organ system toxicity | [50] |
| 4. | Benzophenone: Oxybenzone, avobenzone, and di-oxybenzone. <br> **Used in**: Sunscreen, hair spray <br> **Function**: UV-A rays protection | **The amount used in PCPs**: 3–6% $w/w$ <br> **Major health concerns**: Bioaccumulative, Heptotoxic, Endocrine disruptor, increase risk of skin cancer, irritation and sensitization <br> **Regulatory Status**: <br> • Used as active drug <br> • Monograph available in USP 29 but not in BP 2017, EP 7.0, JP (17th edition), and IP 2018 <br> USFDA and Health Canada accepts up to 6% in their products not more than 0.5% to protect product formulation | [51–53] |
| 5. | Butylated Compounds: Butylatedhydroxyanisole <br> **Used in**: Hair products, make up, sunscreen | **The amount used in PCP**: 0.1% $w/w$ in cream, 0.1% $w/w$ in gel, 0.1% $w/w$ in shampoo, 0.1% $w/w$ in spray <br> **Amount used in other pharmaceutical products**: Buccal Gum—0.21 mg, Nasal Spray—0.1 mg, Oral Capsule—0.4 mg, Oral Tablet—0.81 mg. <br> **Major health concern**: Organ-system toxicity and irritation | [54] |
| 6. | Coal tar <br> **Used in**: Shampoos, soaps, hair dyes, lotion <br> **Function**: Control psoriasis, dandruff | **The amount used in PCP**: 0.5% to 5% (US) <br> **Major health concerns**: Cancer, organ toxicity, bioaccumulation | [55] |

**Table 2.** *Cont.*

| S.No. | Category and Name of the Ingredients, Types of Cosmetic Products and Functions | The Amount Used in PCP, Other Pharmaceutical Products and Major Health Concerns | References |
|---|---|---|---|
| 7. | Ethanolamine: Diethanolamine and Triethanolamine <br> **Used in**: Soaps and hair products <br> Eye makeup <br> **Function**: Emulsifier, pH adjuster | **The amount used in PCP**: ≤5% in products applied for longer time (US) and ≤2.5% in non-rinse off products(EU) <br> **Major health concern**: Carcinogenic/hepatocarcinogenic due to formation of nitrosamine on contamination | [56,57] |
| 8. | Ether alcohol: Phenoxyethanol <br> **Used in**: Moisturizer, sunscreen, lotion, shampoo <br> **Function**: Preservative and stabilizer | **The amount used in PCP**: 1.5% $w/w$ in aerosol, 1% $w/w$ in cream, 0.7% $w/w$ in gel, 1% $w/w$ in lotion <br> **Major health concerns**: Allergies, nervous system effects (infants) | [58,59] |
| 9. | Ether: 1.4-dioxane <br> **Used in**: Hair products <br> **Function**: Impurity | **The amount use in PCP**: There is no maximum potency as it is a by-product of the chemicals used <br> **Major health concern**: Breast cancer, organ toxicity, irritation | [60] |
| 10. | Formaldehyde and formaldehyde-releasing preservatives (imidazolidinyl Urea) [59] <br> **Used in**: Nail polish, hair gel, nail adhesive, shampoo for babies <br> **Function**: Preservatives | **The amount used in PCP**: 0.6% $w/w$ (EU) and 0.28%$w/w$ <br> **Amount used in other pharmaceutical products**: <br> Intramuscular injection—3 mg, intravenous injection—3 mg, subcutaneous injection—3 mg <br> **Major health concerns**: Human carcinogen <br> **Regulatory Status**: <br> • Used as excipient. Monographs available in USP 29 but not in BP 2017, EP 7.0, JP (17th edition) and IP 2018 | [59,61] |
| 11. | **Heavy metals**: Lead, arsenic, mercury, chromium, and cadmium <br> **Used in**: Colorant <br> **Function**: Color additive, impurity | **Amount limit**: 10 ppm are allowed as impurity and 20 ppm in color additive for lead <br> 3 ppm for arsenic <br> 1 ppm for mercury <br> 50 ppm for chromium <br> 3 ppm (impurity), 15 ppm as color additive for cadmium. <br> **Major health concerns**: Neurotoxin, developmental and reproductive toxicity. Cancer, mercury poisoning <br> Respiratory and carcinogenic effects <br> Toxic effects on kidneys and skeletal and respiratory system | [62–64] |
| 12. | Isomeric benzenediol: Resorcinol <br> **Used in**: Hair dyes, shampoo, acne products <br> **Function**: preservative | **The amount used in PCP**: 2% when combined with sulfur <br> **The amount used in other pharmaceutical products**: <br> Anorectal drug—1–3% <br> **Major health concerns**: Endocrine Disruptor, Organ system, toxicity, irritant and sensitizer | [65–67] |

**Table 2.** *Cont.*

| S.No. | Category and Name of the Ingredients, Types of Cosmetic Products and Functions | The Amount Used in PCP, Other Pharmaceutical Products and Major Health Concerns | References |
|---|---|---|---|
| 13. | Isothiazolinones: Methylisothiazolinone (MI) Methylchloro-isothiaz-olinone (CMIT) **Used in**: Hair color/cleaner, sunscreen, makeup remover Body wash, lotion, mascara, detergents **Function**: Preservatives | **The amount used**: 100 ppm for rinse off product 15 ppm for rinse off product 0.05% $w/w$ in cream **Major health concerns**: Toxicity/neurotoxicity and allergies | [68–70] |
| 14. | Octinoxate: Octylmethoxycinnamate **Used in**: Shampoo, sunscreen, lipstick **Function**: UV filter | **The amount used**: 7.5% $w/w$ in sunscreen and lip balm **Major health concerns**: Endocrine disruptor, reproductive organs, and development toxicity | [52,71] |
| 15. | PABA (Para amino benzoic acid) and its derivatives): Padimate O **Used in**: Suncreen **Function**: Absorb UV-B radiation | **The amount used in PCP**: In 2019 FDA restricted use of PABA in OTC sunscreen; prohibited in the cosmetic products (EU). 7% $w/w$ in sunscreen **The amount used in other pharmaceutical products**: 300 to 400 mg daily for the treatment of Peyronie's disease **Major health concerns**: Endocrine disruptor, sensitization (increased risk of skin cancer) | [72–74] |
| 16. | Parabens: Butylparaben, Methylparaben and Propylparaben **Used in**: Shampoos and conditioners **Function**: Preservatives | **The amount used in PCP**: 0.4% $w/w$ cream, 0.15% $w/w$ lotion, 0.18% $w/w$ ointment 0.16% $w/w$ in aerosol, 0.5% $w/w$ in cream, 0.15% $w/w$ in shampoos 5.25% $w/w$ in cream, 0.2% $w/w$ in lotion, 0.03% $w/w$ in shampoo **The amount used in other pharmaceutical products**: Ophthalmic Solution—0.02% $w/w$, oral suspension—8 mg/5 mL, oral tablet coated—0.08 mg Auricular formulation (0.01% $w/w$), Buccal Film (1 mg), and Nasal Drops (26 mg) Oral syrups (100 mg), tablet (0.2% mg), and soft tissue injection (0.01% $w/w$) **Major health concerns**: Endocrine disruptors, developmental and reproductive toxicity | [59,75] |
| 17. | Phenol: Hydroquinone **Used in**: Skin lightener, facial moisturizer **Function**: Decreases melanine, make skin appear whiter | **The amount used in PCP**: 2% $w/w$ OTC, higher in prescription only products **The amount used in other pharmaceutical products**: Vaginal cream—0.02% $w/w$ **Major health concerns**: Increase risk of skin cancer, ochronosis | [76,77] |
| 18. | Polyacrylic aromatic hydrocarbons: naphthalene, chrysene, fluorene, anthracene, phenanthrene and acenaphthene **Used in**: Lotions, cosmetics **Function**: Impurity in petrolatum a moisturizing agent | **The amount used**: 5 ppm (US), petrolatum should be free of any carcinogenic impurities (EU) and 500 ppm as color additives (US) **Major health concern**: Cancer | [78,79] |

**Table 2.** *Cont.*

| S.No. | Category and Name of the Ingredients, Types of Cosmetic Products and Functions | The Amount Used in PCP, Other Pharmaceutical Products and Major Health Concerns | References |
|---|---|---|---|
| 19. | Quaternary ammonium salts: Quaternium-15<br>**Used in**: Hair conditioners, cream, lotions, shaving products<br>**Function**: Preservative | **The amount used in PCP**: 0.2% in beauty products (EU), there is no regulation on quaternion-15 in cosmetic products (US)<br>**The amount used in other pharmaceutical products**:<br>0.02% $w/w$ in cream and 0.1% $w/w$ in cream, augmented (US)<br>**Major health concerns**: Irritation, sensitization | [80] |
| 20. | Salicylate: homosalate<br>**Used in**: Sunscreen, Sun protection<br>**Function**: Convert UV radiation into Infrared radiation | **Amount used**: 15% $w/w$ in sunscreen<br>**Major health concern**: Endocrine disruptor, impact hormones | [52,81] |
| 21. | Titanium dioxide<br>**Used in**: Sunscreen, makeup powder<br>**Function**: UV filter, colorant | **The amount used in PCP**: 25% in sunscreen (EU, US)<br>**Amount limit in food**: 1% of the weight of the food as colorant<br>**Major health concern**: Carcinogen | [82,83] |
| 22. | Triclosan<br>**Used in**: Antibacterial soaps, toothpaste, antiperspirant<br>**Function**: Antibacterial | **Amount use**: Banned for OTC<br>**Major health concern**: Endocrine disruptor, bioaccumulative | [84,85] |

**Table 3.** (a): Safety-based global clinical trials of sunscreen products. (b): Safety-based global clinical trials of baby products. (c): Safety-based global clinical trials of skin care/other products.

| S. No. | Title | Status | Invention | Proof of Concept/Evaluation Parameters | Location | NCT Number |
|---|---|---|---|---|---|---|
| | | | **(a)** | | | |
| 1. | Systemic Absorption evaluation of Sunscreen components [86] | Completed (Phase 1) | **Products (part-1)**:<br>Cream, lotion, and spray<br>**Products (Part-2)**:<br>Lotion, aerosol spray, non-aerosol spray, and pump spray<br>**Excipients (part-1)**:<br>Avobenzone, octocrylene, ecamsule<br>**Excipients (part-2)**:<br>Avobenzone, oxybenzone, octocrylene | Concentration optimization of active drugs | Spaulding Clinical Research, West Bend, WI, USA | NCT03582215 |
| 2. | Safety assessment of a Sunscreen C-Spray on Sport persons [87] | Completed | Coded products | Assessment of erythema as graded on a 5 point scale. | Saint Petersburg, FL, USA | NCT02857478 |
| 3. | Safety assessment of Sunscreen Product [88] | Completed | Coded products | Dermatologist's subjective and objective assessments of potential Adverse events. | Saint Petersburg, FL, USA | NCT02803320 |

**Table 3.** *Cont.*

| S. No. | Title | Status | Invention | Proof of Concept/Evaluation Parameters | Location | NCT Number |
|---|---|---|---|---|---|---|
| | | | **(a)** | | | |
| 4. | Irritation studies of Sunscreen Products in Human Eyes [89] | Completed | Coded products | (1) Macroscopic evaluations for lacrimation, bulbar conjunctiva irritation, palpebral conjunctiva irritation. (2) Subjective assessment of discomfort | Saint Petersburg, FL, USA | NCT02854137 |
| 5. | Phototoxicity study of Sunscreen Products [90] | Completed | Coded products and Drug: Sodium chloride | (1) Primary outcome measures—Skin irritation (2) Secondary outcome measures—Number of adverse episodes as an assessment of safety and tolerability | Pinellas Park, FL, USA | NCT02802930 |
| 6. | Sun Protection Factor (SPF) analysis of Defense Skin Cream [91] | Completed | Physiogel Daily Defense Protective Day Cream Light | Skin pH, transepidermal water loss, intercellular lipid lamellae length are measured. | Pro-derm Institute for Applied Dermatological Research, Schenefeld, Germany | NCT03136107 |
| 7. | Analysis of sun protective product according to sunscreen formulas (Study SR09-15) (P08236) (COMPLETED) (PFA and SPF) [92] | Active, not recruiting | Group discussions | Frequency of sunscreen preference differences between ideal and willingness to pay | Northwestern University, Chicago, IL, USA | NCT01001975 |
| 8. | Repeat insult patch study of Daily Facial Moisturizer and Sunscreen with SPF 50+ (Cetaphil®) on human [93] | Completed | ISO 24444:2010 P3 Standard Sunscreen | Arithmetic mean of individual sun protection factor (SPFi) value are calculated. | GlaxoSmithKline (GSK) Investigational Site, Schenefeld, Schleswig-Holstein, Germany | NCT01892657 |
| 9. | Evaluate the sun protection factor of sunscreen products [94] | Completed | Based of SPF used in three sunscreen lipcare products | Minimum effective concentration has analysed for protected and unprotected sites. | GSK investigational sites, Winston-Salem, North Carolina, USA | NCT05085327 |
| 10. | Assay of SPF and UVA Protection Factor (UVAPF) [95] | Completed | Fresh coconut oil, jojoba oil, almond oil, and ointment of white petrolatum | Product satisfaction of the following test articles A, B, and C: Lindi Skin Soothing Balm (Product A), Lindi Skin Face Serum (Product B), and Lindi Skin Face Wash (Product C). | Northwestern University Department of Dermatology, Chicago, IL, USA | NCT02872246 |
| 11. | Sun Protection Factor (SPF) Efficacy Assay [96] | Completed | Lindi skin care products | Evaluation of water resistant SPF | Union, NJ, USA | NCT02885805 |

**Table 3.** *Cont.*

| S. No. | Title | Status | Invention | Proof of Concept/Evaluation Parameters | Location | NCT Number |
|---|---|---|---|---|---|---|
| | | | **(a)** | | | |
| 12. | Clinical study of general skin care product for analyze their effect on the Structural Strength of the Skin [97] | Completed (Phase 3) | Survey | (1) Determination of sunscreen protection factor (SPF) <br> (2) Determination of ultraviolet A protection factor (PFA) | No Contacts or Locations Provided | NCT03625167 |
| 13. | Clinical assessment of sunscreen products on adult [98] | Completed | Coded products and SPF 15 Control | (1) Sun protection factor (SPF) effectiveness on the skin of human participants before 2 h of water immersion <br> (2) Sun Protection Factor (SPF) effectiveness on the skin of human participants after 2 h of water immersion | Union, NJ, USA | NCT02877511 |
| 14. | Clinical assessment of sunscreen products on women [99] | Completed | Coded products and SPF 15 Control | Skin response to sun exposure according to the Skin Evaluation Response Scale from 0 to 3 | Saint Petersburg, FL, USA | NCT02779270 |
| | | | **(b)** | | | |
| 1. | An evaluation of tolerability study of three wash products in infants [100] | Completed | Test shampoo, Test bath foam and Test head to toe wash | Observation of baseline in Corneometer Values for 8 hrs. | GSK Investigational Site, Valinhos, Brazil | NCT02403999 |
| 2. | Determination of Tolerability of cleanser and moisturizing product with SPF 30 on kids [101] | Completed | Coded products and Standard cleanser soap | Frequency of combined dermatologist score and ophthalmologist score | GSK Investigational Site, Campinas, São Paulo, Brazil | NCT01909713 |
| 3. | Efficacy of Baby Talcum in Prevention of Pruritus Associated With Cast [102] | Recruiting | Treatment with petrolatum | Schirmer test is performed. | ErolOlcokCorumWEducatin and Research Hospital, Corum, Turkey | NCT01017315 |
| 4. | UK Baby Study using a baby products (body wash and lotion regimen) [103] | Completed | Coded products | Skin response to sun exposure according to the skin evaluation response scale from 0 to 3. | Saint Petersburg, FL, USA | NCT03142984 |

**Table 3.** *Cont.*

| S. No. | Title | Status | Invention | Proof of Concept/Evaluation Parameters | Location | NCT Number |
|---|---|---|---|---|---|---|
| | | | **(b)** | | | |
| 5. | An in-vivo study of skin hydration effect using bath product and moisturizers on human skin in atopic dermatitis [104] | Active, not recruiting | Coded products | (1) Change in neonatal skin condition score (2) Change in Transepidermal Water Loss (3) Change in stratum corneum hydration (4) Number of adverse events reported related to investigations products (Phase 1) | Saint Mary's Hospital, Manchester, Machester, UK) | NCT02028546 |
| | | | **(c)** | | | |
| 1. | Irritation studies of skin care Product on human [105] | Completed | Drug: Coppertone and Coded product. | Number of subjects showing growth in papules, erythema, dryness, telangiectasia, tactile exterior roughness, and irritation. | The Education & Research Foundation, Inc. Lynchburg, VA, USA | NCT03841032 |
| 2. | Clinical Study of cosmetic products for the assessment of efficacy and safety for facial line treatment [106] | Completed | Combination Product: Facial cleanser Combination Product: Facial moisturizer Combination Product: Sunscreen | (1) Clinical Efficacy Graded by Griffiths' Scale (2) Objective Tolerability: scores | Ablon Skin Institute Research Center Manhattan Beach, CA, USA | NCT04545970 |
| 3. | Clinical Study of cosmetic products for the assessment of efficacy and safety for Arm Firming [107] | Completed | Topical Body Firming Moisturizer Other: Body Cleanser Other: Sunscreen Other: Placebo Moisturizer | Change from baseline in skin creepiness score, elasticity score, firmness score, sagging score, skin roughness (visual and tactile), overall photo damage score, evenness of skin tone score, evenness of skin redness score and in objective erythema parameters. | Stephens and Associates Richardson, TX, USA | NCT04065035 |
| 4. | A Clinical Study to analyze the Dermal and Ocular Tolerance of cosmetic Facial Serum (Healthy Females With Sensitive Skin) [108] | Completed | Developmental Serum and Physiogel Calming Relief Anti-Redness Serum | Eczema Area Severity Index (EASI) is used. | Stephens & Associates, Inc Richardson, TX, USA | NCT03719742 |
| 5. | A Clinical Study (Controlled) of two different dry skin care moisturizing products [109] | Completed | Coded products | Number of participants with a unit difference of greater than 1 in Symptoms and signs of cutaneous irritation total result from baseline to 21 days of article use | GSK Investigational Site Schenefeld, Schleswig-Holstein, Germany | NCT03640832 |
| 6. | Local cutaneous and ocular tolerance study for assessment of three developmental facial skin-care products [110] | Completed | Serum, Lotion and Cream | Average difference from baseline for 42 days observation in clinical grading of skin dryness | Thomas J. Stephens & Associates, Inc. Colorado Springs, CO, USA | NCT04510103 |

**Table 3.** *Cont.*

| S. No. | Title | Status | Invention | Proof of Concept/Evaluation Parameters | Location | NCT Number |
|---|---|---|---|---|---|---|
| | | | **(c)** | | | |
| 7. | Characterization of a moisturizing cream on human with blemish prone skin [111] | Recruiting | Washout/Standard Cleanser, Test product and Positive control | Local cutaneous tolerance assessment of participants for irritation determined by a dermatologist | Schenefeld, Schleswig-Holstei, Germany | NCT03093181 |
| 8. | Tolerance study of Cosmetic Product on facial acne for one year regular follow-up [112] | Completed | Coded products | Tolerability assessment of test products | GSK Investigational Site, Edinburgh, UK | NCT04301063 |
| 9. | Analysis of Efficacy and Barrier Protection of Two Cosmetic Products [113] | Active, not recruiting | Coded products | (1) Change in acne severity from baseline to 365 days, change in acne lesions counting, change in Cardiff Acne Disability Index questionnaire, change in subject's global assessment (SGA) of acne extremity. (2) Number of acne relapse(s)/flare(s) (3) Change in pilosebaceous follicles status on forehead area | France-Centre de SantéSabouraud—Hôpital Saint Louis, Paris, France; Cabinet Médical, Sèvres, France; Skin Research Centre, Toulouse, France | NCT03629405 |
| 10. | Clinical trial on characterization of the de-pigmentation activity of a cosmetics on pigmented human skin [114] | Completed (Phase 4) | Emulsion and Gel | Surface area of erythema and increased responses of skin while using product | DermIng SRL, Monza, Italy | NCT02204436 |
| 11. | Clinical study of anti-ageing Facial Gel [115] | Completed (Phase 3) | Facial Moisturizer with SPF 50+ | Skin spots image analysis: change from baseline | DermIng SRL, Monza, Italy | NCT01948531 |
| 12. | Cosmetic Dermatology Study [116] | Completed (Phase 4) | Emulsion and gel | Wrinkles profilometry (Ra—micrometers) test is done. | Derming S.r.l. Single Member Company, Monza, MB, Italy | NCT02200471 |
| 13. | Clinical study of cosmetic product to improve the appearance of human skin affliction with low to optimum atopic dermatitis. [117] | Completed | Gynomunal® gel | Patient perception of the use of the device and physician perception of the use of the device | Dermatology Cosmetic Laser Medical Associates of La Jolla, Inc., La Jolla, CA, USA | NCT03268174 |
| 14. | Cosmetics and Pregnancy (PERICOS) [118] | Completed | Device: VeinViewer | Determination of cosmetics by the physical examination on volunteers | Medical Dermatology Associates of Chicago, Chicago, IL, USA | NCT03283189 |

**Table 3.** *Cont.*

| S. No. | Title | Status | Invention | Proof of Concept/Evaluation Parameters | Location | NCT Number |
|---|---|---|---|---|---|---|
| | | | **(c)** | | | |
| 15. | Effect of Eye Make up on Ocular Surface [119] | Completed | Sunscreen and Standard SPF 4 Sunscreen | Outcome Measures-<br>(1) Blistering time on day 28 (buffer time of 2 days) and at Day 56 (buffer time of 2 days)<br>(2) Difference from baseline in epidermal thickness, hydration, and difference starting from baseline in stratum corneum hydration (SCH), observed in arbitrary units (AU) | Clinical Research Center for Hair and Skin Science, Department of Dermatology and Allergy, ChariteUnive Berlin, Germany | NCT04478955 |
| 16. | Epicutaneous Testing of Cosmetics [120] | Unknown (Phase 3) | Diagnostic Test: Schirmer test | Pruritus score, satisfactory score, complication rate and number of antihistamine drugs used are recorded and measured. | Prince of Songkla University, Hatyai, Songkhla, Thailand | NCT03024671 |
| 17. | Epidermal Delivery of Ani-Aging Ingredients [121] | Recruiting | Baby talcum | Number of patients with positive patch test reactions to cosmetics | Inselspital Bern, Bern, Switzerland | NCT01847066 |
| 18. | Determination of whitening cosmetic product for the lightening effect [122] | Unknown | Patch test application | Clinical photographs to determine improvement of appearance | NY Derm LLC, New York City, NY, USA | NCT01249469 |
| 19. | Determination of skin irritation study/Allergic Sensitivity by skin patch test intended to applied natural personal-care products [123] | Unknown | Device: Erbium 2940 plus cosmetics plus Impact Device: Erbium 2940 plus cosmetics | The average decrease of darkness from baseline in target surface after treatment is observed and noted. | National Taiwan University Hospital, Taipei, Taiwan | NCT01816542 |
| 20. | Ocular stinging potential study of shampoo components on human [124] | Unknown | Sunscreen Agents | Eye irritation was assessed by tearing/inflammation/discomfort/post installation effect on eye | Department of Dermatology, Hadassah University Hospital, Jerusalem, Israel | NCT02869113 |
| 21. | Facial Cosmetic Acupuncture on Skin Rejuvenation [125] | Completed | patch tests on healthy skin | Subjective discomfort in the eye, tearing/lacrimation, objective inflammation, post installation eye effects were measured by an ophthalmologist. | Saint Petersburg, FL, USA | NCT01486303 |
| 22. | Hair Care Product Use Among Women of Color [126] | Unknown | Coded products and Standard shampoo mixture (Control) | Criteria for evaluating Moire's topography are used. | Kyung Hee University Hospital at Gangdong, Seoul, Korea | NCT04493892 |
| 23. | Human Photoallergy Test [127] | Recruiting | Procedure: Facial Cosmetic Acupuncture | Change in internal dose of urinary phthalate metabolites is measured. | Community Engagement Core CommunitySpace, NY, USA | NCT02750449 |

**Table 3.** *Cont.*

| S. No. | Title | Status | Invention | Proof of Concept/Evaluation Parameters | Location | NCT Number |
|---|---|---|---|---|---|---|
| | | | **(c)** | | | |
| 24. | Clinical trial of relative effect of single nucleotide polymorphisms (SNP) to Skin Care Product [128] | Completed | Behavioral: Educational Intervention | Potency of skin interaction is measured by 5 point grading scale | Piscataway, NJ, USA | NCT03446079 |
| 25. | Impact of Exposure to Cosmetics on Sensitive Skin (SENSICOS) [129] | Enrolling by invitation | Drug: Sun screening Sport Lotion (Coded products) and Untreated skin | Genetic Profile & Product Response are assessed. | Halcyon Dermatology, Laguna Hills, CA, USA | NCT03958968 |
| 26. | In-Use Test With a Cosmetic Product [130] | Recruiting | Topical Anti Aging Cream | Comparison of cosmetics use between subjects with and without sensitive skin | Laboratory Interaction Neurones Keratinocytes, Brest, France | NCT03252730 |
| 27. | Determination of retinal toxicity of hair color products containing Aromatic Amines (CAPITOX) [131] | Completed | Cosmetic Exposure | Tolerance of the test product on the scalp | SIT Skin Investigation and Technology Hamburg GmbH, Hamburg, Germany | NCT04222387 |
| 28. | Skin irritation study and sensitivity study through patch test of facial moisturizer with SPF 50 (Cetaphil) [132] | Recruiting | Coded products | Percentage of patients with MEKAR retinopathy noticed on OCT-B scan. | Fondation A de Rothschild, Paris, France | NCT01887860 |
| 29. | Prospective Evaluation of Facial Cosmetic Procedures [133] | Completed | Device: OCT-B scan | Area of erythema and edema measured | AMA Laboratories, New City, NY, USA | NCT03460158 |
| 30. | Quaternium-15, Use Test [134] | Recruiting (Phase 4) | Other: Cetaphil Daily Facial Moisturizer SPF 50 | Patient satisfaction with validated FACE-Q survey | University of Pennsylvania Health System, Philadelphia, PA, USA, | NCT00311454 |
| 31. | Repeat Insult Patch Test of Skin Irritation/Sensitization for CetaphilDermacontrol Oil Control Moisturizer SPF 30 [135] | Completed | Drug: T.R.U.E.Test | (1)  Skin irritation measures<br>(2)  Number of adverse episodes as an evaluation of safety and tolerability | Pinellas Park, FL, USA | NCT01887808 |
| 32. | Tissue augmentation study on human for safety analysis of color products [136] | Completed | Coded products and Drug: Sodium chloride | Surface of erythema and edema to test interaction of skin to article | Galderma Laboratories, L.P., Fort Worth, TX, USA | NCT01147172 |

**Table 3.** *Cont.*

| S. No. | Title | Status | Invention | Proof of Concept/Evaluation Parameters | Location | NCT Number |
|---|---|---|---|---|---|---|
| | | | **(c)** | | | |
| 33. | Barrier of skin, biophysical effect and clinical appearance of moisturizer on human dry skin [137] | Completed | Cetaphil Derma Control Oil and Control Moisturizer SPF 30 | (1) Keloid formation at location of administered area. (2) Pigmentation changes at location of administered area. | Vitiligo and Pigmentation Inst of Southern California, Los Angeles, CA, USA; Skin Care Research, Inc., Boca Raton, FL, USA | NCT03093597 |
| 34. | Dermal effect of cosmetic product on cancer patient: A Survey [138] | Completed | Device: Elevess | Observe the appearance of xerosis | University of Arizona, Banner-University Medical Center, Tucson, AZ, USA | NCT00871429 |
| 35. | Clinical study of OTC product dermally [139] | Completed | Coded products | Hydration capacity of skin using corneometer and loss of water trans-epidermally, value of water loss using Tewameter at different regimens of bath product and moisturizer on trans-epidermal. | Skin Center, Faculty of Medicine, SrinkharinwirotUniversity, Bangkok, Thailand | NCT03641430 |
| 36. | Functional activity of skin care regimen on human skin [140] | Not yet Recruiting | Procedure: bathing and moisturizer application | Photoaging (time frame: 1 year) | Cutaneous Translational Research Program, Department of Dermatology, Johns Hopkins University Schoo Baltimore, MD, USA | NCT03497130 |
| 37. | Assessment of facial product by the potency of photosensitivity and photo allergic reaction on healthy volunteers [141] | Completed | Supportive care with OCT product | (1) TCS variation (total clinical score). (2) determination of visual dryness on dermis | Cutaneous Translational Research Program, Baltimore, MD, USA | NCT03183518 |
| 38. | Clinical studies of dermal irritation/skin sensitivity by using facial products [142] | Completed | skin care regimen | Observation of superficial irritation scores and skin irritation score at different time interval | GSK Investigational Site, Valinhos, Brazil | NCT04007159 |
| 39. | Effect of moisturizing creams on the barriers of skin [143] | Completed | micellar cleansing cream and saline solution | Percentage of participants with potential sensitization reactions as assessed by dermatologist on day 40 | GSK Investigational Site, Campinas, São Paulo, Brazil | NCT03804710 |
| 40. | Clinical studies of cleanser by cutaneous/ocular tolerance on healthy women with dermal sensitivity [144] | Completed | Serum, Lotion, Cream and Normal Saline | Loss of water from dermal skin at time interval | GSK Investigational Site, Schenefeld, Schleswig-Holstein, Germany | NCT03172364 |
| 41. | Tolerance and Efficacy Evaluation of 3 Face Creams (FILLER) [145] | Completed (Phase 4) | Micellar cleanser and Micellar foaming cleanser | Cutaneous tolerability based on visual inspection—erythema, edema, dryness, and roughness. | RCTS, Inc., Irving, TX, USA | NCT02063971 |

### 5. Safety Assessment of Excipient Used in Cosmetic/Personal Care Products

In the analysis of risk, safety margin (infertility factors) is determined in the last step of safety assessment. According to WHO, a minimum value of 100 is required for margin of safety (MoS) is required to accept the compound for minimum adverse effect/safety purpose and their equation is shown below [45].

$$Margin\ of\ safety\ =\ \frac{Non\ -\ observed\ adverse\ effect}{Systemic\ exposure\ dose} \tag{1}$$

NOAEL is the maximum amount of dose from which no treatment-related side effects are observed. The *SED* of a personal care product is the predicted amount that enters into the blood circulation and is expressed as mg/kg BW/day. Whereas *SED* is expressed as % of component penetrating the skin as shown in the equation.

$$SED\ =\ \frac{daily\ dermal\ exposure\ \times\ DAp}{60\ \text{kg}\ of\ bw} \tag{2}$$

Cosmetic ingredient review (CIR) was organized by the PCPC (Personal Care Product Council) Expert panel of CIR and constitutes industry representatives, toxicologists, dermatologists, and chemists [29]. The purpose of the meetings is to re-evaluate all the published and unpublished research, industry data, the literature on all chemicals of cosmetics being determined to check the safety of chemicals that are involved in the formulation of cosmetics [146]. Furthermore, based on the voting system, safety evaluations are finalized and concluded and then published in peer-reviewed journals. Documents of guidance are developed and followed for the cosmetics/ personal care products manufacturing and quality assurance unit [3].

The scientific committee on consumer safety is a self-governing body that is capable to suggest the European Commission. By risk assessment process, it also delivers the guidelines for characterizing and analyzing of the safety of cosmetics [147]. International Council of Chemical Associations is responsible to publish guidance on risk assessment of chemicals which govern the comprehensive strategy used to appraise risk factors of chemicals that are incorporated in cosmetics. Guidance documents are published to meet the criteria for the development of safe products and for the betterment of health worldwide [148]. Another framework, the International Fragrance Association (IFRA) was implemented to design and publish the standard safety guidelines for the use of fragrance ingredients and components in a wide range of cosmetic/personal care products. The standard of fragrance components is developed by IFRA with regard to risk assessment strategies that facilitate recently applied level, category of cosmetic item, and exposure potency during application along with scientists of Research Institute for Fragrance Materials [149]. Evaluation and conclusion are appraised by advisory experts including skin specialists, physicians, toxicologists, and other required scientists. Analyzed data that are published in peer review journals and standard criteria of IFRA for fragrance components are also uploaded on the website. The Flavor and Expert Manufacturers Association (FEMA) research scientists rely on a structurally similar group of chemicals as functional materials of its risk assessment strategies for food flavoring compounds [150].

### 6. Conclusions

The human body is exposed to various chemical substances. A substantial amount of hazardous chemicals are being used in the preparation of cosmetic products such as preservatives, perfumery, surfactant, colorant, and other excipients. Most of these chemicals possess toxicity and are known to cause various complications. For human grade products, there is a requirement to follow legal or regulatory guidelines and risk assessment to cover all possible aspects of using chemicals in cosmetics and PCPs. In some cases, if any ingredient exceeds the limits, especially in case of children or pregnant ladies, it can cause permanent damage to the skin, internal organs, as well as the brain. Many of these

ingredients present in body care products can even cause cancer, neural damage, and permanent skin disorders.

According to the authors, the guidelines pertaining to safety should be made stringent and the compliance should be made mandatory. It is a general practice that products are granted marketing authorization on the basis of safety information or data of ingredients. However, there should also be a safety assessment of product as a whole. In the given point of view, the generation of preclinical data becomes mandatory but there are ethical concerns of using animals for the same. It can be addressed by using in vitro tools such as cell line studies, artificially generated skin models etc. The authors also suggest to put pharmacovigilance system in place for cosmetic products. The group is already working in this direction, and a survey study in this regard is also in process.

**Author Contributions:** Conceptualization, M.K. and U.F.; methodology, M.K.; software, M.S.A.; validation, M.A.M. and Z.I.; writing—original draft preparation, M.K.; writing—review and editing, U.F.; M.J.A. visualization, M.J.A.; supervision, M.A.M.; project administration, Z.I.; funding acquisition, M.J.A. All authors have read and agreed to the published version of the manuscript.

**Funding:** This research received no external funding.

**Institutional Review Board Statement:** Not applicable.

**Informed Consent Statement:** Not applicable.

**Data Availability Statement:** Not applicable.

**Acknowledgments:** The authors are highly grateful to the Jamia Hamdard University, New Delhi, India for all its support.

**Conflicts of Interest:** The authors declare no conflict of interest.

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
