# Peer review of "Safety Concern and Regulatory Status of Chemicals Used in Cosmetics and Personal Care Products"

_dermato, doi:10.3390/dermato3020011_

Round 1
Reviewer 1 Report
In my opinion, the submitted manuscript „Safety concern and regulatory status of chemicals used in cosmetics and personal care products” meets aims and scope of „Dermato” Journal and may be accepted after revision.
1. According to the PRISMA guidelines for reviews (http://prisma-statement.org/PRISMAStatement/Checklist.aspx) authors should: „Specify the inclusion and exclusion criteria for the review and how studies were grouped for the syntheses. Specify all databases, registers, websites, organizations, reference lists and other sources searched or consulted to identify studies. Present the full search strategies for all databases, registers and websites, including any filters and limits used. Specify the methods used to decide whether a study met the inclusion criteria of the review, including how many reviewers screened each record and each report retrieved, whether they worked independently, and if applicable, details of automation tools used in the process…” The manuscript does not describe how the literature for this review was collected – it should be added (in the Introduction part, line 91), authors must underline the selective criteria and the restrictions for the literature search.
2. The reference for cosmetics definition in line 39 is needed.
3. Why in table 1, are listed the definition of a cosmetic in the application of Indian and American law, and there is nothing, for example, about the regulations of the European Union countries (or any other countries)?
4. The text on lines 162-164 needs to be checked, it looks like an additional caption to figure 2.
5. Figures 1 and 5 require improvement, the figure or text overlaps the frames in some places.
6. Table 2 requires a bibliographic reference, or each row in the table requires a reference if they have different sources.
7. In my opinion, the authors should summarize the data they presented, express their personal opinions and give possible ways of solving problems (e.g. what should/could be done so that cosmetics will not be considered as dangerous to health?).
Author Response
Comments and Suggestions for Authors
Reviewer 1
In my opinion, the submitted manuscript „Safety concern and regulatory status of chemicals used in cosmetics and personal care products” meets aims and scope of „Dermato” Journal and may be accepted after revision.
- According to the PRISMA guidelines for reviews (http://prisma-statement.org/PRISMAStatement/Checklist.aspx) authors should: „Specify the inclusion and exclusion criteria for the review and how studies were grouped for the syntheses. Specify all databases, registers, websites, organizations, reference lists and other sources searched or consulted to identify studies. Present the full search strategies for all databases, registers and websites, including any filters and limits used. Specify the methods used to decide whether a study met the inclusion criteria of the review, including how many reviewers screened each record and each report retrieved, whether they worked independently, and if applicable, details of automation tools used in the process…” The manuscript does not describe how the literature for this review was collected – it should be added (in the Introductionpart, line 91), authors must underline the selective criteria and the restrictions for the literature search.
Reply: I have compiled the screening criteria as mentioned in figure 1 according to your comment.
- The reference for the cosmetics definition in line 39 is needed.
Reply: I have added the refence in line 39.
- Why in table 1, are listed the definition of a cosmetic in the application of Indian and American law, and there is nothing, for example, about the regulations of the European Union countries (or any other countries)?
Reply: American guidelines are supposed to be gold standards as they are pioneers in defining such guidelines. We haven't listed all stringent regulatory authority's definitions because major one like USFDA, EMA and Japan are synced in the form of ICH. If the reviewer suggests we can include all.
- The text on lines 162-164 needs to be checked, it looks like an additional caption to figure 2.
Reply: yes, I have made the correction.
- Figures 1 and 5 require improvement, the figure or text overlaps the frames in some places.
Reply: I have improved the image quality and resolution.
- Table 2 requires a bibliographic reference, or each row in the table requires a reference if they have different sources.
Reply: I have edited the references according to their source.
- In my opinion, the authors should summarize the data they presented, express their personal opinions and give possible ways of solving problems (e.g. what should/could be done so that cosmetics will not be considered as dangerous to health?).
Reply: We have mentioned in the conclusion.

Reviewer 2 Report
The study: "Safety concern and regulatory status of chemicals used in cosmetics and personal care products" is very interesting study and gives an important contribution to the growing field of cosmetics. Authors included a plethora of information from published articles and clinical trials. Conclusion is too short. Please add more relevant information as a summary of your findings and propose how this information can be used to make the cosmetics more safe.
Author Response
Comments and Suggestions for Authors
Reviewer 2
The study: "Safety concern and regulatory status of chemicals used in cosmetics and personal care products" is very interesting study and gives an important contribution to the growing field of cosmetics. Authors included a plethora of information from published articles and clinical trials. Conclusion is too short. Please add more relevant information as a summary of your findings and propose how this information can be used to make the cosmetics more safe.
Reply: We have made changes in the conclusion.
Thank you
